# A De novo Peptide from a High Throughput Peptide Library Blocks Myosin A -MTIP Complex Formation in *Plasmodium falciparum*

**DOI:** 10.3390/ijms21176158

**Published:** 2020-08-26

**Authors:** Zill e Anam, Nishant Joshi, Sakshi Gupta, Preeti Yadav, Ayushi Chaurasiya, Amandeep Kaur Kahlon, Shikha Kaushik, Manoj Munde, Anand Ranganathan, Shailja Singh

**Affiliations:** 1Special Centre for Molecular Medicine, Jawaharlal Nehru University, New Delhi 110067, India; zillzz85@gmail.com (Z.e.A.); preetiyadav.bms@gmail.com (P.Y.); ayushi.chaurasiya01@gmail.com (A.C.); amangenomics@gmail.com (A.K.K.); shikhakaushik29@gmail.com (S.K.); 2Department of Life Sciences, School of Natural Sciences, Shiv Nadar University, Greater Noida, Uttar Pradesh 201304, India; nj633@snu.edu.in; 3School of Physical Sciences, Jawaharlal Nehru University, New Delhi 110067, India; sakshi2027@gmail.com (S.G.); mundemanoj@gmail.com (M.M.)

**Keywords:** malaria, peptide inhibitor, myosin A, myosin A tail interacting protein (MTIP)

## Abstract

Apicomplexan parasites, through their motor machinery, produce the required propulsive force critical for host cell-entry. The conserved components of this so-called glideosome machinery are myosin A and myosin A Tail Interacting Protein (MTIP). MTIP tethers myosin A to the inner membrane complex of the parasite through 20 amino acid-long C-terminal end of myosin A that makes direct contacts with MTIP, allowing the invasion of *Plasmodium falciparum* in erythrocytes. Here, we discovered through screening a peptide library, a de-novo peptide ZA1 that binds the myosin A tail domain. We demonstrated that ZA1 bound strongly to myosin A tail and was able to disrupt the native myosin A tail MTIP complex both in vitro and in vivo. We then showed that a shortened peptide derived from ZA1, named ZA1S, was able to bind myosin A and block parasite invasion. Overall, our study identified a novel anti-malarial peptide that could be used in combination with other antimalarials for blocking the invasion of *Plasmodium falciparum*.

## 1. Introduction

The burden caused by a malarial parasite—*Plasmodium falciparum*—remains huge despite a recent decline in the number of malaria cases [1]. The non-responsiveness of the parasite to existing therapies calls for an urgent need for discovering new drugs that are parasite-specific. For successful transmission, the parasite must complete its life cycle in each of the sub microenvironments: gut of Anopheles mosquito, hepatocytes, and erythrocytes. Entry in erythrocytes allows asexual reproduction of the parasite, leading to rapid proliferation and an exponential increase in the number of merozoites. It is an essential step in the parasite life cycle and is also an attractive target since the merozoites are in the bloodstream and directly exposed, thereby making them vulnerable to drugs [2].

The encounter of the malarial parasite with host erythrocyte is mediated by sequential and highly dynamic processes involving initial interaction of the egressed merozoite, leading to re-orientation of the apical end, the formation of a tight junction in between the two membranes, followed by the engagement of invasion motor and entry. This is concluded by the shedding of a protein coat, the formation of the parasitophorous vacuole, resealing of red blood cell (RBC) membrane, and echinocytosis [2,3,4]. The process is governed by a large number of parasite proteins that work in a coordinated manner during each sub-stage of invasion. Merozoite surface proteins (MSPs) lead to initial contacts between RBC and parasite membrane [5,6]. Next, the rhoptries and micronemes create a gradient of invasion proteins, resulting in the apical end orientation towards RBC [7]. This is accompanied by a rise in cytosolic Ca^2+^ levels, engagement of erythrocyte binding antigen 175 (EBA 175) with glycophorin A on RBCs [8], and formation of Rhoptry Neck Protein (RON) 2/4/5-Apical Membrane Antigen 1 (AMA1) complex [9,10,11]. This engagement sets the stage for the formation of a tight junction, allowing the motor machinery of the parasite to generate the force by hydrolyzing ATP, marking the entry of the parasite in host RBC. Thus, the glideosome machinery links parasite movement to invasion. Many essential proteins and protein–protein interactions of the machinery occur exclusively in the parasite, are highly conserved across different strains and isolates of the parasite, and are non-redundant, making them strong drug targets [4,9,12]. The actinomysin motor is present in the inner membrane complex of the parasite and composed of actin, myosin A, myosin A tail interacting protein (MTIP), essential light chain (ELC), glideosome-associated proteins—GAP 40, 45, 50, and thrombosponding-related anonymous protein (TRAP). A central player in the glideosome machinery is myosin A.

*P. falciparum* possesses six types of myosin (*Pf* myosin A–F) [13], of which myosin A belongs to class XIV unconventional myosin that is unique to apicomplexa [14]. Pf myosin A is an 818 amino acid protein, comprising distinct motor and light chain binding domain. The motor domain binds actin in an ATP-dependent manner [15,16]. The light chain binding domain binds MTIP [17,18,19] and ELC [20]. The tail domain of myosin A is sufficient for its binding to MTIP [21] and is extremely well conserved across *Plasmodium species* [19]. Inhibitors like cytochalasin that blocks actin polymerization [22,23] and butanedione monoxime (BDM) that target myosin [24,25] have shown to block locomotion in apicomplexa. The pivotal role of myosin A is also well characterized in *Toxoplasma gondii* infection [26] and *Plasmodium* species [19,27], disruption of which leads to the impairment in the invasion. Recent *Pf* myosin A knockout studies have also confirmed its direct role in motility, as well as invasion [28], making it an attractive drug target.

Interaction of MTIP to myosin A via its tail domain immobilizes myosin A to the inner membrane complex (IMC). Myosin A/MTIP complex disruption by urea-based inhibitors [27] and nanobodies [29] have been shown to block the invasion of *Plasmodium*. Other peptide-based inhibitors targeting myosin A or MTIP [19,30,31] have also been shown to be promising. Here, we discovered a de novo peptide binder—ZA1—against the tail domain of myosin A, thereby inhibiting the interaction of myosin A with MTIP. We showed that ZA1 was able to bind myosin A tail and displace MTIP. We also discovered another peptide—ZA1S—from the original de novo peptide ZA1, but shorter in length, that was able to enter the parasite, bind Myosin A tail, and block invasion.

## 2. Results

### 2.1. Validation of the Myosin A tail/MTIP Interaction

*Pf* myosin A/MTIP interacting regions have been mapped by previous studies [17,18]. The interaction of myosin A to MTIP is via the 20 amino acid long C terminal tail of myosin A (Figure 1a) that makes direct contacts with the C terminal domain of MTIP [19,21]. In order to ascertain this interaction in our system, the interacting domains of *Pf* myosin A (798–818 residues) and MTIP (61–204 residues) were codon-optimized and cloned in bacterial two-hybrid vectors—pTRG_nn_ and pBT_nn_—and a two-hybrid assay was set up. We observed a strong interaction between myosin A tail and MTIP, owing to the blue-colored colonies obtained on X-gal indicator plates (Figure 1b). The relative strength of the interaction was quantified using the liquid β-galactosidase assay. A well-documented interaction between *Mycobacterial* proteins—ESAT6 and CFP10—was used as the positive control, while the combination of ESAT6 and the empty vector pBT_nn_ was taken as the negative control (Figure 1c).

Next, an enzyme-linked immunosorbent assay (ELISA-based interaction assay) was performed to test the interaction. Myosin A tail (798–818 residues) was acquired from Genescript (Appendix A), while 17.4 kDa recombinant MTIP-His (61–204 residues) was purified from *E. coli* BL21 (DE3) under native conditions, followed by dialysis against PBS (Appendix A). For ELISA, MyoA tail was coated on the ELISA plates, and MTIP was overlaid. After stringent washing, the binding of MTIP was detected using an anti-MTIP antibody. We found the myosin A/MTIP interaction to be concentration-dependent (Figure 1d). The specificity of the interaction was tested by the ELISA-based interaction of MTIP with the unrelated protein ESAT6. No interaction was found (Appendix A). The interaction of myosin A tail and MTIP was also studied by Isothermal Titration Calorimetry (ITC). The binding of myosin A tail to MTIP showed negative enthalpy (binding enthalpy change ΔH: −4.7 kcal/mol) and positive entropy (TΔS: 3.8 kcal/mol, ΔG = −8.6 kcal/mol), suggesting that the reaction was both enthalpy and entropy-driven. The dissociation constant (K_d_) for myosin A/MTIP was found to be 452 nM. A sigmoidal curve ending near the zero baselines indicated saturation of all binding sites.

### 2.2. Screening Peptide Binder against Myosin A Tail Domain

The driving force that allows the parasite to invade the host cell is generated by the cooperative action of the components of the actin-myosin motor. Myosin A is a central component of the glideosome machinery, and recent studies have highlighted it as a potential drug target [28]. The interaction of myosin A with MTIP allows tethering/anchorage of myosin A into the inner membrane complex; targeting this interaction, therefore, can help in blocking the entry of the parasite in the host erythrocyte. Hence, we sought to identify a peptide binder of myosin A tail—the region that fits inside the MTIP domain—thereby not allowing it to bind MTIP. We screened a dicodon polypeptide synthetic library against the myosin A tail domain. The bacterial two-hybrid analysis was performed with myosin A tail as the bait and the dicodon shuffled library as prey. In this system, the target is fused to the N-terminal domain of RNA Polymerase in the pTRG vector, and the bait protein is fused to λ repressor that binds DNA on the operator upstream to the promoter. When the bait and target interact, RNA Polymerase is recruited to the promoter and stabilized along with the transcriptional machinery, leading to transcriptional activation of the β-galactosidase reporter gene, resulting in blue colored colonies [32,33]. The selection of colonies is based on blue-white screening that is later confirmed by multiple rounds of re-patching, re-transformation, re-co-transformation, and sequencing of the interacting peptide obtained from the dicodon library [34,35,36,37].

Of all the positive interactions, after the above-mentioned screening, we obtained four peptides; ZA1, ZA2, ZA3, and ZA4 (Appendix A). Out of these, a de novo 80 amino acid long peptide binder named ZA1 was identified and annotated (Figure 2a). The strength of myosin A tail/ZA1 interaction was assessed by liquid β-galactosidase assay (Figure 2b). *Mycobacterial* proteins—ESAT6 and CFP10 (that are known to bind strongly with each other)—were used as the positive control, and ESAT6 and empty vector pBT_nn_ were taken as the negative control. We have previously identified peptide binders against crucial proteins of *Mycobacterium tuberculosis* and *Plasmodium falciparum* using a similar approach [34,35]. The biophysical features like isoelectric point and grand average of hydropathicity (GRAVY) value were studied for stability (Figure 2c). ZA1 also showed the maximum intensity of blue color on the bacterial two-hybrid selection plate (Appendix A).

### 2.3. ZA1 Interacted with Myosin A Tail In Vitro

The interaction between myosin A tail and ZA1 was confirmed through multiple protein–protein interaction studies. A 10.3 kDa recombinant ZA1-His was purified from *E. coli* BL21 (DE3) under denaturing conditions and then refolded by removing denaturant gradually by dialysis (Figure 2c). An ELISA-based interaction assay was performed to test the interaction of myosin A tail with its de novo peptide binder. For this, myosin A tail was coated on the ELSIA plates, followed by overlaying of ZA1. After stringent washing, the binding of overlaid ZA1 protein was detected using an anti-His antibody. We found a positive interaction between myosin A and ZA1 that was concentration-dependent (Figure 3a). The specificity of this interaction was determined by testing the ELISA-based interaction of ZA1 with an unrelated protein ESAT6, with which no interaction was found (Appendix A).

In order to characterize further the interaction of ZA1 with myosin A tail, ITC studies were performed. Figure 3b shows the typical binding isotherm for titration of myosin A into ZA1. A sigmoidal curve ending near the zero baseline indicated the saturation of all binding sites (Figure 3b). The data were fitted using a single site model. The binding interaction resulted in 1:1 stoichiometry, suggesting that one molecule of myosin A bound with one molecule of ZA1. The binding was observed to be enthalpically opposed (ΔH: +2.29 kcal/mol) and entropically favorable (TΔS: +10.66 kcal/mol, ΔG = −8.37 kcal/mol), resulting in a strong binding affinity with a K_d_, of 719 nM. In order to rule out the possibility of heat changes due to myosin A, a parallel control experiment was run in which myosin A was sequentially added into the buffer; it showed no heat changes (Appendix A). Overall, these results confirmed the interaction of novel polypeptide ZA1 with myosin A tail.

### 2.4. ZA1 Disrupted Myosin A tail-MTIP Complex In Vivo

In order to study the effect of ZA1 on myosin A/MTIP complex formation in vivo, we used the bacterial three-hybrid approach, previously developed in our laboratory [36]. This system allowed us to transform the original blue reporter strain carrying MyoA-pTRG and MTIP-pBT plasmids (blue because of the positive and strong myosin A/MTIP interaction) with a third compatible plasmid ZA1-pMTSA. The expression of ZA1 could be tightly regulated through the arabinose-based induction system. Upon transforming the double-transformed cells with ZA1-pMTSA and growing them on X-gal indicator plate in the presence of arabinose, ZA1 was expressed, and the color of the colonies changed from blue to white, indicating the disruption of myosin A/MTIP complex. ZA1-pMTSA triple-transformed cells grown on X-gal plates in the absence of arabinose remained blue (Figure 4a). A control plasmid carrying only the vector pMTSA was also transformed and exhibited no effect of arabinose on myosin A/MTIP complex formation, giving blue-colored colonies both in the presence and absence of arabinose.

The quantitative nature of disruption of the complex was studied by inducing the triple transformed cells with a gradient of arabinose (0–4 mM) and measuring the strength of interaction of myosin A /MTIP complex at different concentrations of ZA1 by liquid β-galactosidase assay. A reduction in the strength of interaction was observed with increasing amounts of ZA1. The specificity of the interaction was observed with no change in the strength of interaction of myosin A/MTIP when transformed with the pMTSA vector only. A gradual increase in the levels of ZA1 upon increasing arabinose was confirmed by Western blot analysis by detecting the amount of ZA1 expressed at various arabinose concentrations (Figure 4b).

### 2.5. ZA1 Disrupted Myosin A tail-MTIP Complex In Vitro

In order to ascertain the binding of ZA1 with myosin A tail in the presence of MTIP, we performed competitive binding experiments utilizing ITC and ELISA. Myosin A tail was coated on ELISA plates, followed by the application of an equimolar ratio of MTIP to allow myosin A/MTIP complex formation. The complex was overlaid with ZA1 in increasing concentrations, and after stringent washing, the amount of MTIP bound to myosin A tail was detected using a specific antibody. The binding of myosin A to MTIP was found to be conversely dependent on ZA1 concentrations. The amount of MTIP bound to myosin A tail reduced with increasing ZA1, confirming the preferential binding of myosin A to ZA1 in the presence of MTIP (Figure 4c).

The ability of ZA1 to disrupt the myosin A/MTIP complex was further tested by competitive binding analysis using ITC. Here, the MTIP-bound myosin A was titrated into ZA1, the hypothesis being that if the myosin A/MTIP complex is able to bind ZA1, it would demonstrate the ability of ZA1 to interfere with the myosin A-MTIP complex. Figure 4d shows that MTIP-bound myosin bound to ZA1 but with a weaker affinity (K_d_: 57 µM) compared to free myosin A binding to ZA1. The interactions were driven by unfavorable enthalpy (ΔH: 60.13 kcal/mol) and favorable entropy (TΔS: 65.85 kcal/mol; ΔG = −5.72 kcal/mol). Thus, 80 times decrease in the binding strength clearly indicated competition between ZA1 and MTIP for myosin A. However, the absence of complete elimination of the binding strength suggested two possibilities: (1) ZA1 bound partially to an MTIP site and thus interfered with myosin-MTIP complex, or (2) ZA1 had an additional binding site on myosin A in addition to its MTIP binding site. The graph showed an endothermic binding curve (in contrast to exothermic binding curve of myosin A/MTIP complex), with positive enthalpy and entropy in agreement to binding characteristics observed for myosin A/ZA1 binding (Figure 4d).

### 2.6. A Shorter Peptide ZA1S Bound Myosin A Tail and Inhibited the Invasion of P. falciparum

To understand better the residue to residue interaction between myosin A tail and ZA1, in silico studies were performed. The myosin A/ZA1 complex interacted via eleven hydrogen bonds, and the binding was found to be favorable with a minimum binding energy of −586.4 kcal/mol (Figure 5a,b, Appendix A). Interestingly, most polar contacts lay along a stretch of amino acids from aspartate 10 to glutamate 18. This encouraged us to study the binding of myosin A tail with a shortened ZA1:ZA1S (DIDIDIMHE). Docking studies between myosin A tail and ZA1S revealed a favorable interaction with a minimum binding energy −447.3 kCal/mol and eleven hydrogen bonds (Figure 5c,d).

Next, the direct interaction of ZA1S with myosin A tail was tested by Dot-Far Western analysis. For this, ZA1S peptide was immobilized on a strip of nitrocellulose membrane, followed by an overlay with biotinylated myosin A. After stringent washing, the bound myosin A was detected using an anti-biotin antibody. Myosin A was detected in the spot corresponding to ZA1S and not BSA, thus confirming the interaction (Figure 6a). The interaction of MyoA/ZA1S was also validated by ELISA. Myosin A tail showed significant binding to ZA1S (Figure 6b). These results confirmed that ZA1S bound to myosin A tail. To examine the effect of ZA1S peptide on *P. falciparum’s* 3D7 growth, we performed a growth inhibition assay, taking untreated assay wells as positive control and treating parasites in late trophozoite/schizont stage with increasing concentrations of ZA1S. This stage of the parasite life cycle was chosen due to the known role of myosin A-MTIP complex during invasion [18,19]. We found a concentration-dependent increase in invasion inhibition, though there was no change in the parasite morphology (Figure 6c).

## 3. Discussion

Myosins are motor proteins that couple chemical energy to mechanical energy by ATP hydrolysis, thereby generating the force-allowing movement. Myosin A belongs to an atypical class of myosins and has been shown to be crucial for motility and invasion of *Plasmodium* [28] and other apicomplexa [26].

In this report, we first validated the *Pf* myosin A/MTIP interaction through a bacterial two-hybrid assay and found it to be exceptionally strong. The myosin A/MTIP complex formation has been shown to be essential for the invasion of *Plasmodium* [19], and the disruption of this complex abrogates parasite entry in the host erythrocyte. To be sure, the tail domain of myosin A (residues 799–818) has been shown to be sufficient to bind MTIP [21,38,39,40] and generate the inward propelling force. This region is highly conserved across *Plasmodium* species (*P. yoelli, P. falciparum, P. vivax, P. reichenowi, P. knowlesi, P. bergei, P. chabaudi*) and strains [19]. Hence, we chose the myosin A tail domain as a target and sought to inhibit the myosin A/MTIP complex interaction.

By using a part-rational approach that allowed the synthesis of a myriad of proteins in the test-tube, we discovered a de novo peptide—ZA1—that bound strongly to the myosin A tail domain that, as described previously, fits inside the MTIP pocket. This approach allowed directed evolution by codon shuffling, resulting in a large variety of proteins, all distinct in sequence, secondary structure, and binding capacities against the bait myosin A tail. While the entire *Pf* myosin A protein is more or less neutral, the tail domain is highly basic with an isoelectric point of 12.02. Since charge-based interactions are the primary driving force for protein–protein interactions [38,41,42], we used a preferentially skewed negatively-charged library (DI-EL di-codon library) in order to find a peptide binder of myosin A tail. The bacterial two-hybrid system allowed the screening of a multitude of library members in a quasi-high through-put manner, helping us to ultimately fish out ZA1. Using a similar approach, we have previously unearthed potent peptide binders against crucial proteins of *Mtb* and *Pfal* [34,35,43]. We quantitatively assessed the strength of the interaction between the bait (myosin A tail) and prey (di-codon binder) and found it to be significant. We substantiated myosin A tail/ZA1 interaction by various in vitro protein–protein interaction techniques like ELISA and ITC. Next, we confirmed both in vivo and in vitro whether the peptide ZA1 could disrupt the myosin A tail/MTIP pre-formed complex.

Myosin A/MTIP is part of a larger gliding motility complex present in the parasite’s IMC [3,4]. Due to its size, ZA1 cannot enter into the parasite directly. Hence, with the aim to find precise residues involved in the binding so as to truncate ZA1, bioinformatics studies were performed. A nine amino acid long peptide ZA1S that docked well with myosin A tail domain and interacted through hydrogen bonds was further identified. The acidic and basic nature of ZA1S and myosin A tail, respectively, hinted at a charge-based interaction. The interaction of myosin A tail with ZA1S was validated in vitro by Far Western Dot blot assay and ELISA. Finally, the effect of the shorter de novo peptide on the growth and invasion of *P. falciparum* was studied in culture. The shorter ZA1S peptide abrogated the invasion of *P. falciparum* in RBCs (Figure 7). However, we did not find a complete disruption of myosin A/MTIP complex so as to completely inhibit parasite invasion in both in vivo (bacterial three-hybrid) and *Pfal* growth inhibition assay. This could be due to the inherently tight nature and closed conformation of the complex [39]. To conclude, our study reported the discovery of novel peptide-based inhibitors—ZA1 and ZA1S—that could be used as templates and scaffolds for developing better therapeutics and, in particular, pave the way for peptidomimetic studies against defined targets, thereby allowing parasite-specific targeting. The peptide inhibitors can also be used in combination with other drugs for better anti-malarial efficacy.

## 4. Materials and Methods

### 4.1. Construction of De Novo Peptide Libraries

Dicodon library was constructed, as described previously [43,44]. A 100 ng of each DNA hexamer or dicodon was used to set up a 20 µL ligation reaction with 7.5% polyethylene glycol. The mixture was gently heated to 55 °C, followed by slow cooling to 4 °C and incubation at 4 °C for another 24 h. To this ligation mixture, 100 pmol of 5′-phosphorylated double-stranded hairpins that had previously been self-annealed was added. The ligation mix was incubated at 16 °C for 12 h. Next, ligated dicodon DNA was precipitated using the phenol:chloroform precipitation protocol and re-suspended in nuclease-free water. The precipitated DNA was subjected to *Xba*I digestion at 37 °C for 4 h, and 1 µL of this digested DNA was used as the template for PCR amplification, using HP2P primers that served both as forward and reverse primers. The elution of PCR fragments (100–500 bp) was carried out using the Diethylaminoethyle (DEAE) membrane, followed by the precipitation of phenol:chloroform. The resulting DNA fragments were used as inserts for cloning into *Sna*BI-cut dephosphorylated vectors—pTRG_nn_ and pBT_nn_.

### 4.2. Cloning of Myosin A Tail and MTIP in Bacterial Two-Hybrid Vectors

The interacting regions of *Pf* myosin A (798–818 amino acids) and MTIP (61–265 amino acids) were codon-optimized and synthesized in pUC57 through Genescript (Appendix A). MyoA tail and MTIP inserts were prepared by PCR amplification using gene-specific primers (MyoA For/pTRG Rev, MTIP For/pBT Rev), followed by restriction digestion and blunt-end cloning in pTRG and pBT. The positive clones were confirmed by sequencing.

### 4.3. Bacterial Two-Hybrid Studies

Bacterial two-hybrid experiments were performed, as described by the manufacturer (Stratagene, San Diego, CA, USA). Briefly, equal amounts of both pTRG and pBT plasmids (250 ng each) were used to co-transform R1 reporter cells. The cells were plated on X-gal indicator plates, containing kanamycin (50 µg/mL), tetracycline (12.5 µg/mL), chloramphenicol (30 µg/mL), X-gal (80 µg/mL), X-gal inhibitor—2-phenylethyl-β-D-thiogalactoside (200 µM), isopropyl β-D-1-thiogalactopyranoside (25 µM), and incubated at 30 °C for 48 h. The blue-colored colonies indicated a positive interaction. A previously known interaction of *Mtb* proteins—ESAT6 and CFP10 [45]—that has also been established by the bacterial two-hybrid system [37] and yields blue-colored colonies upon co-transformation was used as a positive control; ESAT6/empty pBT_nn_, which yielded white-colored colonies, was used as a negative control. For library screening, MyoA-pTRG and the negatively-charged DIEL library cloned in pBT were used to set up the bacterial two-hybrid assay. Initial screening was done by visual blue-white selection. Plasmids from blue colonies were isolated, segregated, and used to re-co-transform the R1 strain in order to confirm the interaction. The nucleotide sequence of the novel dicodon partner was obtained through sequencing.

### 4.4. Beta Galactosidase Assay

To confirm the interaction between the proteins and to quantitatively assess the strength of the interaction, the expression of a β-galactosidase reporter enzyme was measured by a colorimetric assay, as described previously [46]. Statistical significance was tested using a two-tailed unpaired Student’s *t*-test with Welch correction by comparing all values to the negative control.

### 4.5. Cloning, Expression, and Purification of MTIP

For the cloning of MTIP in BlaI-cut pET28a vector, the insert was prepared through restriction digestion using *Sna*BI, followed by the ligation in *Sna*BI-cut dephosphorylated vectors. The ligation mixtures were used to transform R1 competent cells. This was followed by the screening of clones using gene and vector-specific primers (MTIP For/pET Rev).

For the expression of the protein, MTIP-pET 28a plasmid was used to transform BL21 (DE3) cells, and the protein induction was carried out with 0.6 mM Isopropyl β-D-1-thiogalactopyranoside (IPTG) for 3 h at 37 °C. MTIP protein expressed with a C-terminal hexa-Histidine tag was obtained in the supernatant. The cell pellet was re-suspended in phosphate buffer saline pH 7.4 and sonicated till a clear solution was obtained. The solution was subsequently centrifuged at 13,000 rpm for an hour to obtain a clear cell lysate. The lysate was allowed to bind Ni-NTA beads overnight at 4 °C, and the protein was subsequently purified by slowly increasing imidazole concentration in PBS. The eluted fractions were run on 15% SDS-PAGE, and the fractions having high concentrations of the protein were pooled and subsequently dialyzed against PBS pH 7.4. The purified protein was run on 15% SDS-PAGE and confirmed by Western blot analysis using an anti-MTIP antibody (1:5000 dilution, Appendix A).

### 4.6. Cloning, Expression, and Purification of ZA1

The ZA1 insert was prepared through the amplification by HP2P primers of the original ZA1-pBT library clone and ligated in the pMTSA vector by blunt-end cloning. The orientation of positive clones was confirmed by sequencing. ZA1-pMTSA clone was used for the expression of ZA1 and bacterial three-hybrid studies.

For functional characterization and in vitro protein–protein interaction studies, ZA1 protein was prepared. First, the expression of ZA1 was confirmed in *E. coli* BL21 (DE3) by arabinose-based induction. The protein was found in inclusion bodies. For large scale purification, BL21 cells carrying ZA1-pMTSA plasmid were grown in liquid culture overnight at 37 °C with streptomycin (50 µg/uL). The secondary culture was inoculated with 1% inoculum and grown till mid-log phase in the presence of streptomycin and induced with 1% arabinose for 3 h at 37 °C. The cell pellet was sonicated in a re-suspension buffer (10 mM Tris HCl, 150 mM NaCl pH 7.4) in the presence of Protease Inhibitor Cocktail, (PIC) (1X) and phenylmethylsulfonyl floride (PMSF) (2 mM) until a clear solution was obtained. The pellet was solubilized in the solubilization buffer (10 mM Tris HCl, 300 mM NaCl, 8 M Urea, 10 mM imidazole pH 7.4), and the resulting supernatant containing the protein was bound to Ni-NTA resin at room temperature. The bound protein was eluted by the imidazole gradient in the solubilization buffer. The eluted fractions were run on Tris-Tricine PAGE, and the fractions carrying the protein were pooled and dialyzed in 10 mM Tris HCl, 150 mM NaCl pH 7.4, along with a gradual decrease in urea so as to enable refolding of the protein. The purified protein was run on Tris-Tricine PAGE and confirmed by Western blot analysis using an anti-His antibody (Figure 2c).

### 4.7. ELISA-Based Assay

ELISA-based protein–protein interaction assay was carried out, as described previously [47], with slight modifications. Briefly, 500 ng MyoA peptide was coated in 0.1 M carbonate/bicarbonate buffer (SPL Maxisorb ELISA plates) for 2 h at 37 °C and blocked at 4 °C overnight in blocking buffer (5% skimmed milk in PBST). Subsequently, the plates were washed three times with washing buffer (1 × PBS, 0.05% Tween-20) and overlaid with increasing concentrations of the second interacting protein partner (0, 0.1, 0.25, 0.5, 1, 2.5 µM) in test wells. In the negative control wells, an unrelated mycobacterial protein ESAT6 was overlaid at the same concentrations. The proteins were allowed to bind for 2 h at 37 °C after which the wells were washed. Next, the overlaid bound protein was detected using a specific primary antibody (rabbit anti-MTIP antibody 1:10,000 dilution, mouse anti-His antibody 1:2000 dilution) in all wells except antibody control wells. A non-specific antibody anti-CFP10 (1:5000 dilution) was used in these wells. After binding for 2 h and washing, the horseradish peroxidase-conjugated secondary antibody (anti-rabbit and anti-mouse 1:5000 dilution) was allowed to bind to the washed plates for 2 h at 37 °C. The ELISA plates were washed, followed by the addition of 3,3′,5,5′-Tetramethylbenzidine (TMB) substrate and incubated at 37 °C for 15–30 min until color development. The reaction was stopped by 2 N H_2_SO_4_, and the absorbance was recorded at 450 nm. All wells were put up in triplicates. ELISA between ZA1S and myosin A tail was performed in a similar manner by coating ZA1S and overlaying it with MyoA tail, with the detection carried out using the anti-biotin antibody.

### 4.8. Isothermal Calorimetric Assay

ITC experiments were performed using ITC_200_ Microcalorimeter (Malvern, UK) at 25 °C in phosphate buffer. The cell and syringe volumes were kept 310 µL and 40 µL, respectively. The syringe was loaded with 100 µM MyoA tail for MyoA/MTIP interaction study and 150 µM for MyoA/ZA1 interaction. The sample cell was loaded with 10 µM each of MTIP and ZA1. MTIP and ZA1 were titrated using MyoA tail in injections of 2 µL each except for the first injection that was 0.4 µL. In between each titration, a 150-s interval for stirring was kept to ensure that the signal returns to baseline. Throughout the experiment, constant stirring at 300 rpm was maintained to ensure proper mixing of reaction components. For both experiments, a corresponding buffer control was also set up in which PBS was taken in the syringe. The enthalpy changes of titrations were measured, and the baseline values were subtracted from the test samples. The data analysis was carried out using Origin software and curve-fitted to obtain stoichiometry (N), dissociation constant (K_d_), and enthalpy change (ƍH).

### 4.9. Bacterial Three-Hybrid Studies

For bacterial three-hybrid experiments, MyoA-pTRG and MTIP-pBT were used to co-transform R1 competent cells. The resulting co-transformants were checked for the presence of both plasmids by PCR using gene and vector-specific primers (MyoA For/pTRG Rev, MTIP For/pBT Rev). The resulting blue colonies were used to prepare competent cells that were subsequently transformed with ZA1-pMTSA or control pMTSA plasmid and plated on X-gal indicator plates, both in the presence or absence of L-arabinose (2% *w*/*v*). The plates were incubated at 30 °C for 48 h. The color of the colonies indicated the formation or disruption of myosin A/MTIP complex. The background and validation of bacterial three-hybrid assay have been described previously [36].

### 4.10. Arabinose Gradient Liquid Beta Galactosidase Assay

An arabinose gradient assay in liquid culture was performed, as described previously [46]. Briefly, the mid-log phase culture was induced with varying concentrations of L-arabinose (0–4 mM) for 3 h at 37 °C. The cells were pelleted, and the levels of β-galactosidase were measured [36]. *E. coli* BL21 (DE3) cells were induced with varying concentrations of L-arabinose (as described in the arabinose gradient assay). Whole-cell lysates of a 10 mL culture were used to analyze the expression of ZA1 using an anti-His antibody (1:2000 dilution).

### 4.11. Competitive ELISA

For ELISA-based inhibition of the interaction between MyoA/MTIP complex using ZA1, 500 ng peptide was coated on Maxisorb ELISA plates, followed by overnight blocking at 4 °C. The bound peptide was overlaid with MTIP (1 µM) for 2 h at 37 °C. The plates were washed with wash buffer (1 × PBS, 0.05% Tween-20) thrice and overlaid with increasing concentrations of ZA1 (0 µM, 0.5 µM, 1 µM, 2 µM, 5 µM) for 2 h at 37 °C. The plates were again washed, and the amount of bound MTIP was detected using an anti-MTIP antibody (1:10,000 dilution), followed by HRP-conjugated secondary antibody. Optical density was taken at 450 nm. All wells were put in triplicates.

### 4.12. Competitive Isothermal Calorimetric Assay

For the competitive ITC experiment, a pre-bound complex of MyoA-MTIP in an equal molar ratio (150 µM each) was taken in the syringe. The complex was used to titrate ZA1 (10 µM) present in the cell. As a control, the pre-bound MyoA-MTIP complex was titrated in buffer only. All parameters were kept identical to the one-on-one ITC experiment. Origin software was used for data analysis and for calculating binding parameters like stoichiometry (N), binding constant (K_d_), and enthalpy change (ƍH).

### 4.13. In Silico Studies

X-ray diffraction structure of MyoA was obtained from the protein data bank (PDB ID: 1QAC) [48]. For ab initio modeling of ZA1 and ZA1S, the I-tasser tool was employed [49]. The quality of the resulting model was validated using the Ramachandran plot analysis tool PROCHECK [50]. For protein–protein docking of ZA1 and ZA1S with MyoA tail, the ClusPro tool was used with default settings [51]. For analysis of the docking results, Pymol, PDBSUM, and Ligplot+ tools were used [52,53,54].

### 4.14. Far Western-Dot Blot Assay

For Far-Western Dot-blot assay, 5 µg each of ZA1S peptide, BSA, and non-specific peptide (N′- *p*SDNG*p*SGDD -C′) was immobilized on the strip. Subsequent to the drying of the membrane, it was blocked overnight at 4 °C, following which the second peptide myosin A was overlaid in 5 µg/mL of binding solution (1% BSA/PBST) on the strip for 2 h at room temperature. The strip was washed stringently, and the bound myosin A peptide was detected using an anti-biotin antibody (1:100 dilution) and HRP-conjugated secondary antibody.

### 4.15. Parasite Culture

*P. falciparum* 3D7 strain was thawed and cultured in vitro in human O+ erythrocytes at 4% hematocrit using the method described previously [55]. Briefly, parasites were cultured in medium containing RPMI 1640 (Invitrogen, Carlsbad, CA, USA), supplemented with 2 g/L NaHCO_3_ (Sigma, St. Louis, MO, USA), 5 g/L albumax (Invitrogen), 50 mg/L hypoxanthine (Sigma), and 10 µg/mL gentamicin sulfate (Invitrogen, Carlsbad, CA, USA) at a pH of 7.2 under mixed gas condition (5% CO_2_, 5% O_2_, 95% N_2_) at 37 °C.

### 4.16. Ethics Statement

*P. falciparum* culture and related experiments were done in accordance with the guidlines of Institutional Bio Safety Committee (IBSC) of Jawaharlal Nehru University, New Delhi.

### 4.17. Growth Inhibition Assay (GIA)

For the growth inhibition assay, 36–38 h’ parasites were purified by percoll (GE Healthcare, Chicago, IL, USA) density gradient centrifugation. The assay was put on the late trophozoite stage in a 24-well plate in quadruplets in a total 500 µL volume/well. Initial parasitemia and hematocrit were adjusted to 1% and 2%, respectively. ZA1S peptide was added at a final concentration of 10 µM, 20 µM, and 40 µM. Control wells without any peptide or inhibitor were set to validate the normal growth rate of the parasite. After 24 h, a smear was made and fixed in methanol, followed by staining with 10% Giemsa (Sigma, St. Louis, MO, USA) for 10 min. The percent of infected erythrocytes containing different stages of the parasite was counted using a light microscope under 100X oil-immersion. The percent of growth inhibition was calculated using the following formula, and data were plotted graphically.
% Growth inhibition = (Control-treated)/Control × 100

## Figures and Tables

**Figure 1 ijms-21-06158-f001:**
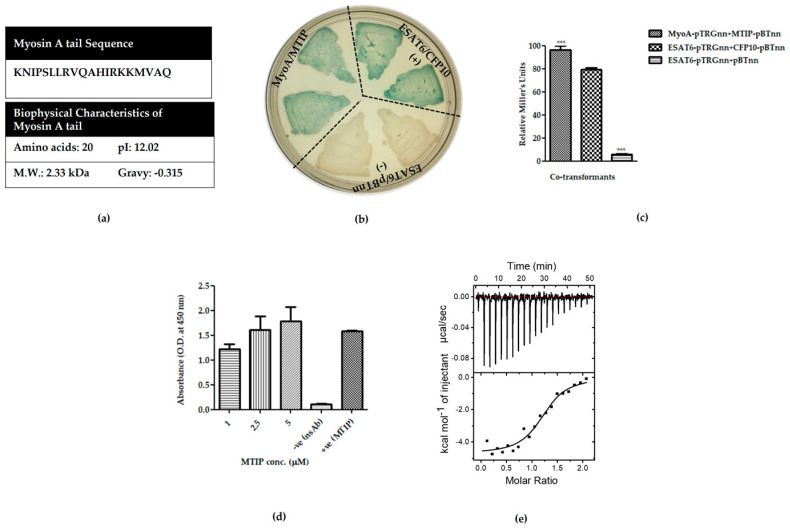
Validation of interaction between myosin A tail and myosin A tail interacting protein (MTIP). (**a**) Amino acid sequence and biophysical characteristics of myosin A tail. Expasy’s Protpam tool (https://web.expasy.org/protparam/) was used to find the molecular weight, isoelectric point, and hydropathicity of myosin A tail. (**b**) X-gal indicator plates, showing liquid patching of bacterial two-hybrid assay between myosin A/MTIP (native interaction). (**c**) Liquid β-galactosidase assay to quantitate the relative enzyme activity (Miller Units) for co-transformant pairs: Myosin A-pTRGnn versus MTIP- pBT 96.14 ± 3.45, ESAT6-pTRGnn versus CFP10-pBTnn 79.17 ± 1.67 (positive control), ESAT6-pTRGnn versus empty pBTnn 5.73 ± 0.62 (negative control). All streaks are labeled to represent genes cloned in pTRGnn/pBTnn. ESAT6-pTRGnn/CFP10-pBTnn and ESAT6-pTRGnn/empty pBTnn are the positive and negative controls, respectively. The graph is the average of three independent assays, with error bars showing standard deviation. All values were tested for significance using a two-tailed unpaired student’s *t*-test with Welch’s correction *** *p* < 0.0005. Validation of interaction between MyoA tail and MTIP protein by (**d**) enzyme-linked immunosorbent assay (ELISA) and (**e**) isothermal titration calorimetry (ITC) studies. For ELISA, myosin A was coated on ELISA plates. MTIP was overlaid at increasing concentrations and detected by anti-MTIP antibody. MTIP coated on plates was used as positive control. Negative control for the ELISA experiment is shown in Appendix A. ELISA was performed in triplicates, and each bar represents mean ± standard deviation shown by error bars. ITC data upper panel depicts the raw calorimetric values, denoting the amount of heat (exothermic peaks) produced following the sequential injection of myosin A into MTIP. The amount of heat generated per injection as a function of the molar ratio of myosin to MTIP is depicted in the bottom panel. K_d_ of myosin A/MTIP interaction was found to be 452 nM. Buffer control is shown in Appendix A.

**Figure 2 ijms-21-06158-f002:**
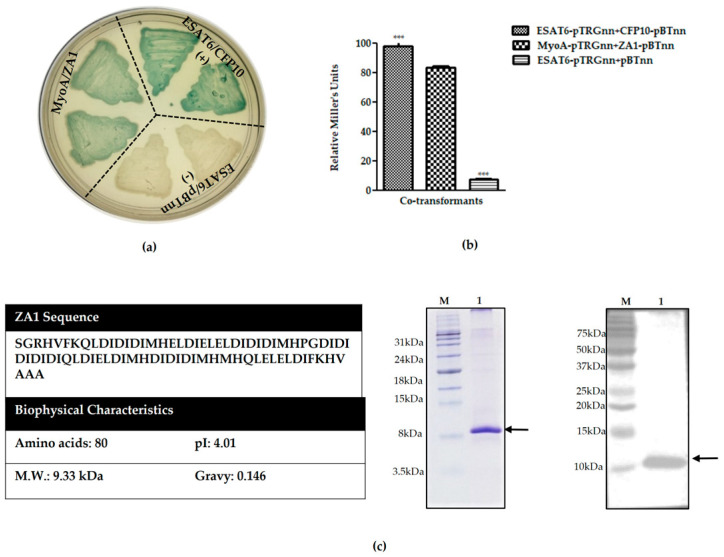
Identification of a de novo peptide that binds myosin A tail. (**a**) X-gal indicator plates, showing liquid patching of bacterial two-hybrid assay between myosin A/ZA1. (**b**) Liquid β-galactosidase assay to quantitate the relative enzyme activity (Miller units) for co-transformant pairs: Relative Miller units: Myosin A-pTRGnn versus ZA1-pBTnn 83.48 ± 0.82, ESAT6-pTRGnn versus CFP10-pBTnn 98.02 ± 2.07 (positive control), ESAT6-pTRGnn versus empty pBTnn 7.10 ± 0.77 (negative control). All streaks are labeled to represent genes cloned in pTRGnn/pBTnn. ESAT6-pTRGnn/CFP10-pBTnn and ESAT6-pTRGnn/empty pBTnn are the positive and negative controls, respectively. The graph is the average of three independent assays, with error bars showing standard deviation. All values were tested for significance using a two-tailed unpaired student’s *t*-test with Welch’s correction *** *p* < 0.0001. (**c**) Amino acid sequence and biophysical characteristics of ZA1 polypeptide were identified from the de novo codon-shuffled library. Expasy’s Protpam tool (https://web.expasy.org/protparam/) was used to find the molecular weight, isoelectric point, and hydropathicity of ZA1. The purification of His-tagged ZA1 was done by Ni-NTA column chromatography under denaturing conditions, followed by a refolding step. Coomassie-stained Tris tricine gel. Lane M: Marker, Lane 1: Purified C-terminal His-tagged ZA1 (10.33 kDa). Western blot of ZA1 using anti-His antibody.

**Figure 3 ijms-21-06158-f003:**
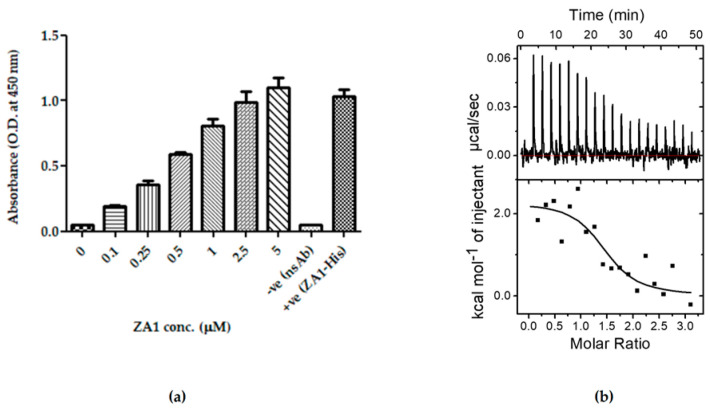
In vitro protein–protein interaction studies between myosin A tail and ZA1 peptide. Validation of interaction between MyoA and ZA1 by (**a**) ELISA-based interaction assay and (**b**) ITC. ZA1 coated on plates was used as positive control. Negative control for the ELISA experiment is shown in Appendix A. The ELISA for all interactions was performed in triplicates, and each bar represents the mean ± standard deviation shown by error bars. ITC data: the upper panel depicts the raw calorimetric values, denoting the amount of heat (endothermic peaks) produced following the sequential injection of myosin A into ZA1. The amount of heat generated per injection as a function of the molar ratio of myosin A to ZA1 is depicted in the lower panel. K_d_ of myosin A/ZA1 interaction was found to be 719 nM. Buffer controls are shown in Appendix A. Final ITC data were obtained by subtracting the data with background heats.

**Figure 4 ijms-21-06158-f004:**
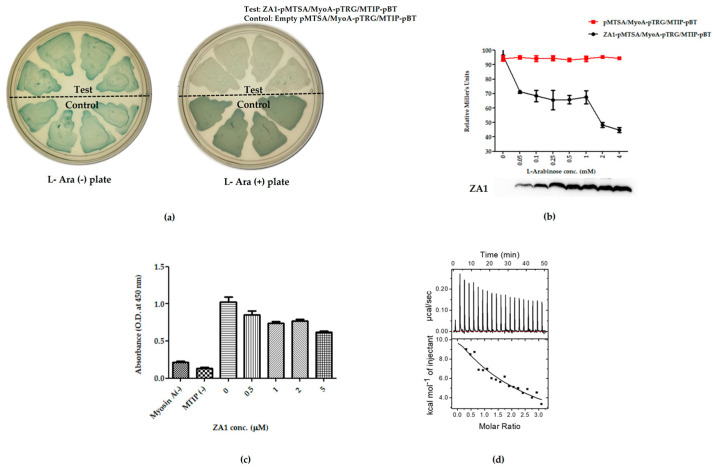
Disruption of myosin A/MTIP interaction by the de novo peptide binder. (**a**) Bacterial three-hybrid assay to study the disruption of myosin A/MTIP complex by ZA1 in the absence and presence of L-arabinose. *E. coli* triple co-transformant, containing ZA1-pMTSA, myosin A-pTRGnn, and MTIP- pBT (test), and triple co-transformant, containing empty pMTSA, myosin A-pTRGnn, and MTIP- pBT (control), on LB agar without and with L-arabinose. (**b**) L-arabinose gradient liquid β-galactosidase assay. Relative Miller units of triple co-transformant pairs: empty pMTSA, myosin A-pTRGnn, MTIP- pBT (red line) and ZA1-pMTSA, myosin A-pTRGnn, MTIP- pBT (black line) plotted against a range of L-arabinose concentrations. The graph is an average of three independent assays, and the standard deviation is represented by error bars. Western blot of *E. coli* whole-cell lysates to analyze the expression of ZA1 with increasing concentrations of L-arabinose using anti-His antibody, followed by the detection using HRP-conjugated secondary antibody. (**c**) ELISA-based inhibition of the interaction between myosin A and MTIP using the inhibitor ZA1. Myosin A was coated on ELISA plates, followed by the overlaying of MTIP. Next, ZA1 protein was overlaid at the indicated concentrations, followed by the detection of MTIP. As negative controls, in a subset of wells, myosin A and MTIP were not coated. (**d**) Isothermal titration calorimetry competitive binding for the titration of pre-bound myosin A-MTIP complex to ZA1. With each injection of the complex, from the syringe into the cell, heat changes due to myosin A-ZA1 interaction and is plotted against time (upper panel). The amount of heat generated as a function of the molar ratio of myosin to ZA1 is represented (lower panel). Control for the competitive binding experiments is shown in Appendix A.

**Figure 5 ijms-21-06158-f005:**
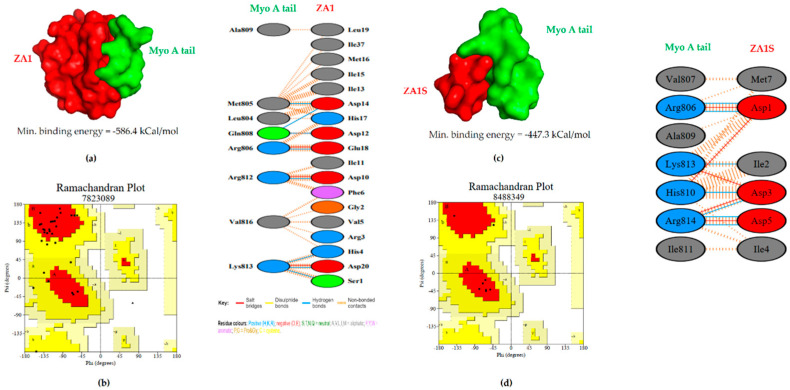
In silico studies for the identification of a shorter peptide: (**a**) The 3D structure of ZA1 (red) interacting with myosin A tail (green) obtained through Pymol. The space-filling model and pictorial representation of residues involved in the interaction. (**b**) Ramachandran plot of Myo A tail/ZA1 structure derived from homology modeling (black dots: residues in the favorable region). (**c**) The 3D structure of ZA1S (red) interacting with myosin A tail (green) obtained through Pymol. The space-filling model and hydrogen bond interface are shown. (**d**) Ramachandran plot of Myo A tail/ZA1S structure derived from homology modeling (black dots: residues in the favorable region).

**Figure 6 ijms-21-06158-f006:**
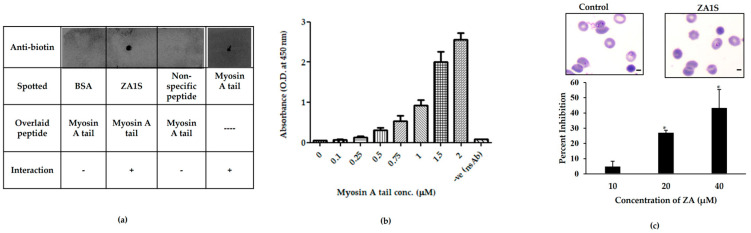
Interaction studies for myosin A tail/ZA1S and effect of ZA1S on *P. falciparum*’s 3D7 growth. (**a**) Far Western-Dot blot assay and (**b**) ELISA to ascertain the interaction between myosin A tail and ZA1S. For Far Western, 5 µg each of ZA1S peptide, BSA, and the non-specific peptide was immobilized on a strip, and the strip was incubated with 5 µg/mL biotinylated myosin A tail, followed by the detection using anti-biotin antibody. Myosin A peptide spotted directly on the membrane was used to check for the antibody. For ELISA, the indicated concentrations of myosin A tail were overlaid on immobilized ZA1S, and the interaction was detected by anti-biotin antibody after stringent washing. (**c**) *Pf*3D7 parasites in smears, representing control and treated sets at 24 hpi (Scale bars: 5 µm). Staging in control and treated sets of 1000 erythrocytes (infected + uninfected) at different concentrations of ZA1S.

**Figure 7 ijms-21-06158-f007:**
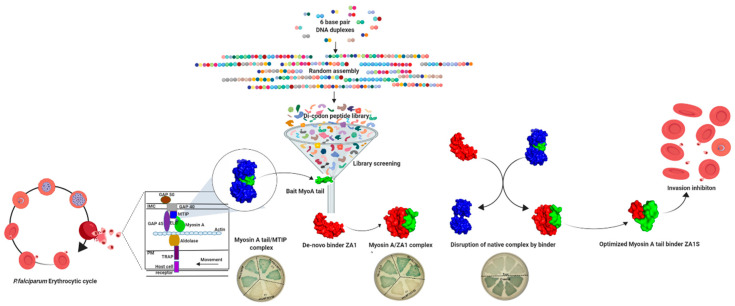
Proposed model for the disruption of myosin A tail/MTIP complex. With the aim to disrupt the crucial myosin A tail-MTIP interaction of the *P. falciparum* motor complex, we found a peptide inhibitor—ZA1—against myosin A tail from a de novo peptide library. ZA1 could potently bind to myosin A tail and disrupt myosin A tail-MTIP interaction. ZA1 was further shortened to ZA1S, allowing the entry of a peptide in the inner membrane complex of the parasite. ZA1S interacted with myosin A tail and inhibited the invasion of *P. falciparum*.

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
