# Peer review of "A De novo Peptide from a High Throughput Peptide Library Blocks Myosin A -MTIP Complex Formation in Plasmodium falciparum"

_ijms, 2020, doi:10.3390/ijms21176158_

Round 1
Reviewer 1 Report
This research is quite interesting and revealing.
Some abbreviations in the abstract and introductory sections needed to be detailed.
Just a few questions to the authors
Based on the outcome of this study, is there a possibility of diversity in the action of this de novo protein in parasite motility and invasion with respect to behavioural characteristics of different plasmodium strains?
Could there be also a possibility of a mutative process that can also interfere in its activity as well?
Reviewer 2 Report
Authors present nice piece of work which can surely impact the malaria research. I have some following concerns in this MS:
- Author used abbreviation in this MS should be expanded when it write first time.(e.g. ELC, BDM etc).
- Author could explain more how they decide or choose peptide from library.
- How author decide the concentration of ZA1 used in growth inhibition assay. I think the timing of ZA1 addition does matter.
